



# On the representativeness of the ground-based lidar observations for satellite calibration/validation—the example of the archipelago of Cabo Verde

Athena Augusta Floutsi[1], Konstantinos Rizos[2], Dimitri Trapon[1], Ronny Engelmann[1],
Dietrich Althausen[1], Eleni Marinou[2], Peristera Paschou[2], Julian Hofer[1], Emmanouil Proestakis[2],
Henriette Gebauer[1], Annett Skupin[1], Albert Ansmann[1], Thorsten Fehr[3], Timon Hummel[4],
Rob Koopman[3], Vassilis Amiridis[2], Ulla Wandinger[1], and Holger Baars[1]

[1]Leibniz Institute for Tropospheric Research (TROPOS), Leipzig, Germany
[2]IAASARS, National Observatory of Athens, Athens, Greece
[3]European Space Agency (ESA), ESTEC, Noordwijk, The Netherlands
[4]European Space Agency (ESA), ESRIN, Frascati, Italy

**Correspondence:** Athena Augusta Floutsi (floutsi@tropos.de)

**Abstract.** Ground-based lidar stations play a vital role in the validation of spaceborne lidar products. While ground-based measurements have a high temporal resolution, they have limited spatial coverage, which potentially imposes implications for the Calibration and Validation (Cal/Val) of the satellite products. Therefore, in this study, we assess the representativeness of a remote ground-based ACTRIS (Aerosol, Clouds, and Trace Gases Research Infrastructure) station, in Mindelo, Cabo Verde by

utilizing the continuous observations of a ground-based PollyNET multiwavelength polarization Raman lidar. This station was selected since Cabo Verde has been a key location for the validation of two recent Earth Explorer missions of the European Space Agency (ESA), namely Aeolus and the Earth Cloud, Aerosol and Radiation Explorer (EarthCARE). The islands are located in the Atlantic Ocean, in the outflow region of the African continent with frequent dust outbreaks, but also smoke advection and, thus, along with the local (marine) boundary layer provide an excellent atmospheric laboratory. Continuous,

vertically-resolved aerosol measurements are being conducted with the state-of-the-art multiwavelength polarization Raman lidar Polly$^{XT}$ at Mindelo since June 2021. Based on these observations and in combination with the LIVAS (LIdar climatology of Vertical Aerosol Structure for space-based lidar simulation studies) products available at different radii around Mindelo, a statistical analysis of the optical properties was performed to evaluate the representativeness of the station in the context of aerosol profiling Cal/Val activities. Additionally, three case studies, focusing on different distances from the ground-based

station, have been closely examined for a more complete and detailed comparison. Our study results indicate that overall the ground-based station in Mindelo can be considered conditionally representative. According to the monthly analysis, at altitudes where the lofted aerosol (dust) layers occur, lidar observations were very representative for radii up to 300 km around the island, while the boundary-layer characteristics varied. Case studies confirmed the long-term results and revealed that lidar observations of lofted aerosol layers can be representative for radii up to 100 km around Mindelo and at the same time

highlighted the importance of spatiotemporal homogeneity of the target. From our findings and especially for the Cabo Verde region, we conclude that it is better to use monthly averaged aerosol profiles for the validation of spaceborne profiles over long



times rather than using single overpasses, as representativeness cannot be guaranteed for the latter without additional measures. Thus, using fixed radii around a certain ground site (as e.g., the frequently used 100 km) for validation activities seems to be inappropriate for profile-to-profile comparison without any further considerations. However, we show in our case studies that if

representativeness can be guaranteed, also single-profile validation is possible and has its own valuable potential. Additionally, the proposed study can serve as a calibration/validation tool for the remote sensing facilities of the European Aerosol Research Lidar Network (EARLINET).

## 1 Introduction

The Calibration and Validation (Cal/Val) activities of spaceborne laser profilers for aerosol and cloud products are essential and, often, quite challenging. In particular, the validation of the calibrated and geolocated measurements (usually referred to as Level 1 data) and of the geophysical parameters related to, e.g., aerosol or clouds (Level 2 data) is crucial to ensure a high-quality dataset from any spaceborne platform. Some commonly encountered challenges are the spatiotemporal scales of the targeted features, the resolution of the spaceborne products and the co-location of the spaceborne and the suborbital

instrumentation (Amiridis et al., 2025, Ch. 2).

CALIOP (Cloud-Aerosol Lidar with Orthogonal Polarization), an elastic-backscatter lidar (532 and 1064 nm) onboard NASA's (National Aeronautics and Space Administration) CALIPSO (Cloud-Aerosol Lidar and Infrared Pathfinder Satellite Observations) satellite, has provided information on the vertical distribution and the optical and microphysical properties of aerosols and clouds (Winker et al., 2009) from 2006 until 2023. The validation of the CALIOP products was of great

importance for the production of a high-quality dataset, especially since CALIOP was not able to perform direct extinction measurements. The lidar ratio (extinction-to-backscatter ratio) had to be assumed to enable the retrieval of the backscatter and extinction coefficients from the attenuated backscatter signals. An aerosol typing scheme, tailored to the CALIOP needs was developed (Omar et al., 2005, 2009; Kim et al., 2018; Tackett et al., 2023) and, regardless of the several quality control procedures in place (Winker et al., 2009), the accuracy of the extinction retrievals was dependent on this typing scheme. Validation

of CALIOP's products was therefore necessary and it was performed by means of direct comparisons with ground-based and airborne measurements.

McGill et al. (2007) performed an initial validation of the CALIPSO Level 1 and 2 products by comparing the spaceborne lidar data with data from CPL (Cloud Physics Lidar), a mobile lidar operating at 1064, 532 and 355 nm onboard the high-altitude NASA ER-2 aircraft, following the suborbital track of CALIPSO. Results showed that the CALIPSO-derived

attenuated backscatter profiles agreed well with the ones from CPL, thus confirming that CALIOP was well calibrated and that the algorithms were performing as expected. Good agreement was found for other CALIPSO products, including aerosol layer detection. Throughout the mission's lifetime, collocated underflights of the NASA Langley Research Center airborne high-





spectral-resolution lidar (HSRL) took place to assess CALIOP's calibration accuracy and to ensure high data quality (Rogers et al., 2011; Kar et al., 2018; Vaughan et al., 2019).

Ground-based systems were also used as part of the validation efforts for CALIPSO, as shown in Mamouri et al. (2009), where the CALIPSO Level 1 attenuated backscatter coefficient profiles were validated using co-located observations performed with a ground-based lidar in Athens, Greece. Wu et al. (2011) performed ground-based lidar measurements in Hefei and the measured attenuated backscatter (at 532 and 1064 nm) and volume depolarization ratio profiles (532 nm) were compared with the ones acquired by CALIPSO. Both studies added valuable information regarding the quality of the CALIPSO products. In

addition, ground-based lidar networks such as the European Aerosol Research Lidar Network (EARLINET, Pappalardo et al., 2014) contributed to the Cal/Val activities through coordinated measurements (Mona et al., 2009; Pappalardo et al., 2010; Papagiannopoulos et al., 2016).

Aeolus, equipped with the 355 nm HSRL ALADIN (Atmospheric LAser Doppler INstrument), was the first spaceborne Doppler wind lidar (Stoffelen et al., 2006). The mission was launched in 2018 by the European Space Agency (ESA) and

within a lifetime of almost five years, ALADIN was established as the first spaceborne lidar able to directly measure extinction profiles and aerosol optical properties as spin-off products (Flament et al., 2021; Baars et al., 2021). Within the framework of the Aeolus Cal/Val, the Joint Aeolus Tropical Atlantic Campaign (JATAC) was organized by ESA and NASA. The campaign comprised of several components, deployed at Cabo Verde (at two-month phases, June and September of 2021 and 2022) and at the U.S. Virgin Islands (2021). Ground-based, aircraft and balloon measurements were conducted, targeting different

objectives, such as the assessment of the quality of the Aeolus products (e.g., Lux et al., 2022; Witschas et al., 2022b) and the validation the several wind- and aerosol-related products (e.g., Borne et al., 2024; Paschou et al., 2025; Trapon et al., 2025). Independent validation efforts were also performed at various locations around the globe, focusing mainly (but not only) on wind products, which compared to aerosol-related products are a more straightforward validation target (e.g., Witschas et al., 2022a; Abril-Gago et al., 2023; Baars et al., 2023; Gkikas et al., 2023; Ratynski et al., 2023).

Apart from the validation of Aeolus data, JATAC primar goals included the study of tropical storms and cyclone formation in the Tropical Atlantic, the study of the interaction between dust particles, wind and clouds, as well as the preparation for future Earth Explorer missions, such as EarthCARE (Earth Cloud, Aerosol and Radiation Explorer; Wehr et al., 2023). In this study, focus is given on the ground-based component of JATAC, which involved several institutes including the Leibniz Institute for Tropospheric Research (TROPOS), the National Observatory of Athens (NOA), the National Institute for Research and

Development for Optoelectronics (INOE), and the National Research Council of Italy - IMAA (CNR-IMAA). This ground-based component, named ASKOS (Marinou et al., 2023), was conducted in Mindelo, on the island of São Vicente, Cabo Verde and the operations included remote-sensing measurements from a complete aerosol and cloud remote sensing facility of ACTRIS (Aerosol, Clouds, and Trace Gases Research Infrastructure; Laj et al., 2024) and in situ measurements from a light aircraft and an unmanned aerial vehicle (UAV).

The ACTRIS facility located at the Ocean Science Center Mindelo (OSCM) and the instrumentation which was available during the ASKOS campaign are both depicted in Fig. 1. The multiwavelength polarization Raman lidar Polly$^{XT}$ (Baars et al., 2016; Engelmann et al., 2016) has been measuring the vertical distribution and optical properties of aerosol continuously since



June 2021. The rest of the instrumentation included a scanning Doppler wind lidar (HALO), a microwave radiometer, a cloud radar (part of ESA's 94-GHz Miniature Network for EarthCARE Reference Measurements- FRM4Radar), a disdrometer, and a
sun photometer (part of the Aerosol Robotic Network- AERONET, Holben et al., 1998). ESA's reference lidar for the Cal/Val of Aeolus, eVe (Paschou et al., 2022), was also measuring during the intensive ASKOS months (not part of the ACTRIS facility). The complete ASKOS dataset (Amiridis et al., 2023), which includes data from multiple instruments optimized for a synergistic usage, has been used mainly for the validation of the aerosol products derived from Aeolus (Paschou et al., 2025; Rizos et al., 2025), but also for wind data assimilation (Georgiou et al., 2023).

JATAC and ASKOS did not only serve as a Cal/Val experiment for Aeolus and supported scientific advances on the interaction of wind, dust and clouds, but provided valuable lessons for ESA's next atmospheric mission, EarthCARE. EarthCARE, a joint mission of ESA and the Japanese Aerospace Exploration Agency (JAXA), was launched in May 2024. EarthCARE's highly sophisticated and complex payload includes an ATmospheric LIDar (ATLID), a Cloud Profiling Radar (CPR), a Multi-Spectral Imager (MSI) and a Broad-Band Radiometer (BBR; Illingworth et al., 2015; Wehr et al., 2023). Optimized for syner-
gistic usage, products include, among others, profiles of clouds, aerosols and precipitation along with co-located radiative flux measurements (Wehr et al., 2023; Eisinger et al., 2024).

Shortly after EarthCARE's launch, from 10 August to 30 September 2024, several campaigns took place on the Cabo Verde islands, Barbados and all across the Atlantic Ocean, under the umbrella of the ORCESTRA (Organized Convection and EarthCARE Studies over the Tropical Atlantic, https://orcestra-campaign.org/orcestra.html, last access: 25 September
2025) project. TROPOS joined ORCESTRA, with a dedicated sub-campaign named CLARINET (CLoud and Aerosol Remote sensing for EarThcare, https://orcestra-campaign.org/clarinet.html, last access: 25 September 2025). The observations collected from the ACTRIS remote sensing facility supported other ORCESTRA sub-campaigns and are currently analysed and used to validate EarthCARE measurements.

Regardless of the several campaigns designed to assist and organize the different Cal/Val activities from the community, to
our knowledge, the degree to which ground-based observations can accurately reflect the atmospheric conditions over a larger scale (e.g., over a few kilometers away) has not been studied in detail so far. Spatial representativeness is especially important for aerosol that can exhibit high spatial heterogeneity and, depending on the aerosol type and meteorology, can have short residence time in the atmosphere.

The profound importance of the Cabo Verde islands on Cal/Val activities, due to the islands being located in the outflow
region of the African continent, along with the open issue of spatial representativeness motivated this study. In the following section, the data used to assess the representativeness of the ground-based Polly[XT] measurements are presented, in addition to the study domain and the methodology. The main findings of the study, i.e., the monthly comparisons between the ground-based and spaceborne optical profiles are presented in Sect. 3. In the same section, three case studies are analysed to examine in detail the issue of spatial inhomogeneity and to reveal potential averaging-induced biases. The concluding remarks and an
outlook are presented in Sect. 4.





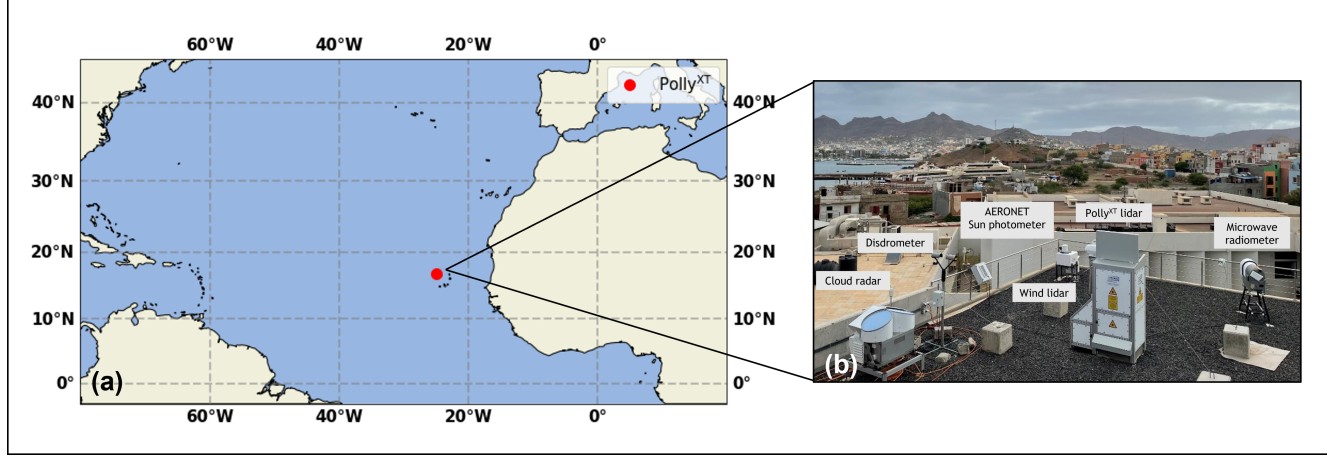

**Figure 1.** (a) Location of the Polly^XT lidar system (red dot) and (b) complete instrumentation of the ACTRIS station at OSCM, Mindelo, Cabo Verde (image source: Holger Baars).

## 2 Data and Methodology

### 2.1 Ground-based and spaceborne datasets

#### Polly^XT

The aerosol remote sensing component (Fig. 1) of the ACTRIS Observational Platform CVAO (Cabo Verde Atmospheric Observatory) includes a state-of-the-art Polly^XT multiwavelength polarization Raman lidar (Baars et al., 2016; Engelmann et al., 2016) among other instruments. As every Polly^XT lidar, this lidar system is also part of the voluntary, scientific, global lidar network PollyNET (https://polly.tropos.de/, last access: 25 September 2025).

Polly^XT utilizes a Nd:YAG laser which emits light at three different wavelengths, 355, 532 and 1064 nm, while the receiver consists of 15 channels, which enables measurements of elastic (355, 532 and 1064 nm) and inelastic backscatter (387, 607 and 1058 nm for aerosols and 407 nm for water vapor) and the depolarization state of the backscatter light (at 355, 532 and 1064 nm). A near-range telescope allows the detection of backscattered light at 355, 387, 532 and 607 nm from about 60–80 m above ground level (a.g.l.). The vertical resolution of the acquired data is 7.5 m and the temporal resolution is 30 s (Engelmann et al., 2016). The multiwavelength capabilities of the Polly^XT lidar systems allow for comparisons with all spaceborne lidars (i.e., CALIOP, ALADIN, ATLID). An in-depth description of the specific lidar system located at Mindelo, together with a discussion of the uncertainties associated with the aerosol optical properties is provided in Gebauer et al. (2024).

The vertically-resolved aerosol optical properties are derived automatically by the PollyNET Processing Chain (PPC; Klamt et al., 2024). This automatic lidar calibration and processing tool, tailored for the PollyNET lidar systems, provides among others vertical profiles of optical properties in near-real-time (NRT). Quicklooks of NRT products can be found at polly.tropos.de (last access: 25 September 2025).



The lidar data considered in this study were collected during the intense campaign periods of ASKOS in 2021 and 2022 in Mindelo, Cabo Verde. Figure 2 shows the aerosol-related Polly[XT] measurements during the ASKOS intensive measurement periods, which are September 2021, June 2022 and September 2022. June 2021 is not considered due to the fact that the instruments were being set up only towards the end of the month because of COVID-19 restrictions at this time, leading to limited measurement availability. By combining the information from the attenuated backscatter signals (Fig. 2a, c, e) with the

volume depolarization ratio (Fig. 2b, d, f), the aerosol conditions above Mindelo can be quickly assessed. In all three periods, lofted aerosol layers occurring at altitudes up to almost 6 km were frequently observed. The elevated attenuated backscatter and volume depolarization ratio values are indicative for non-spherical scatterers, i.e., desert dust aerosol (e.g., Floutsi et al., 2023). Below the lofted dust layers, a marine boundary layer (MBL) was extending up to altitudes of approximately 1 km. The MBL was mostly dust-free, since the observed depolarization ratio is low, indicating the presence of non-depolarizing spherical

particles. The presence of liquid-water and mixed-phase clouds within the MBL was rather frequent during all three months, as indicated by the high values of the attenuated backscatter coefficient and the complete attenuation of the signal above the cloud base (Fig. 2a, c, e).

     In September 2021, three periods with different atmospheric conditions are clearly visible (Fig. 2a, b). Between the 8 and 13 September 2021, a very homogeneous dust layering was observed at altitudes up to 5.5 km. During the period between

the 14 and 18 September, complex horizontal and vertical dust structures were observed, accompanied with high AOD values (not shown here). These layers contained aerosol mixtures of dust and pollution. From 20 September onwards until the end of September 2021, the atmospheric conditions at Mindelo were influenced by the volcanic eruption of Cumbre Vieja at La Palma, Canary Islands, Spain (Gebauer et al., 2024). In the sulfate-dominated planetary boundary layer (PBL), the particle extinction coefficient and lidar ratio were particularly high.

As anticipated, several dust events were observed during both months of the ASKOS 2022 (June and September), which have been also investigated in terms of optical properties in Gebauer et al. (2025). The AOD during those events was high, reaching values up to 0.7, while during the last dust event of September 2022 (21–25 September) the AOD reached values up to 1.

## LIVAS

LIVAS (LIdar climatology of Vertical Aerosol Structure for space-based lidar simulation studies) is a Climate Data Record (CDR) of global, 3-D, multiwavelength aerosol and cloud optical properties (Amiridis et al., 2013, 2015; Marinou et al., 2017) funded by ESA. The LIVAS database is developed based on CALIPSO observations at 532 and 1064 nm and includes the wavelength-converted aerosol optical properties from 532 nm to the LIVAS wavelengths, i.e., 355, 1570 and 2050 nm. The spectral conversion is performed by utilizing the backscatter- and extinction-related Ångström exponent (Å, either ground-

based derived from EARLINET (Pappalardo et al., 2014), or from optical models), a quantity that is aerosol-type-dependent (Floutsi et al., 2023). LIVAS product, include the pure dust and total-aerosol backscatter and extinction coefficients (at several wavelengths), as well as the particle linear depolarization ratio at different vertical smoothing lengths. The mass concentration





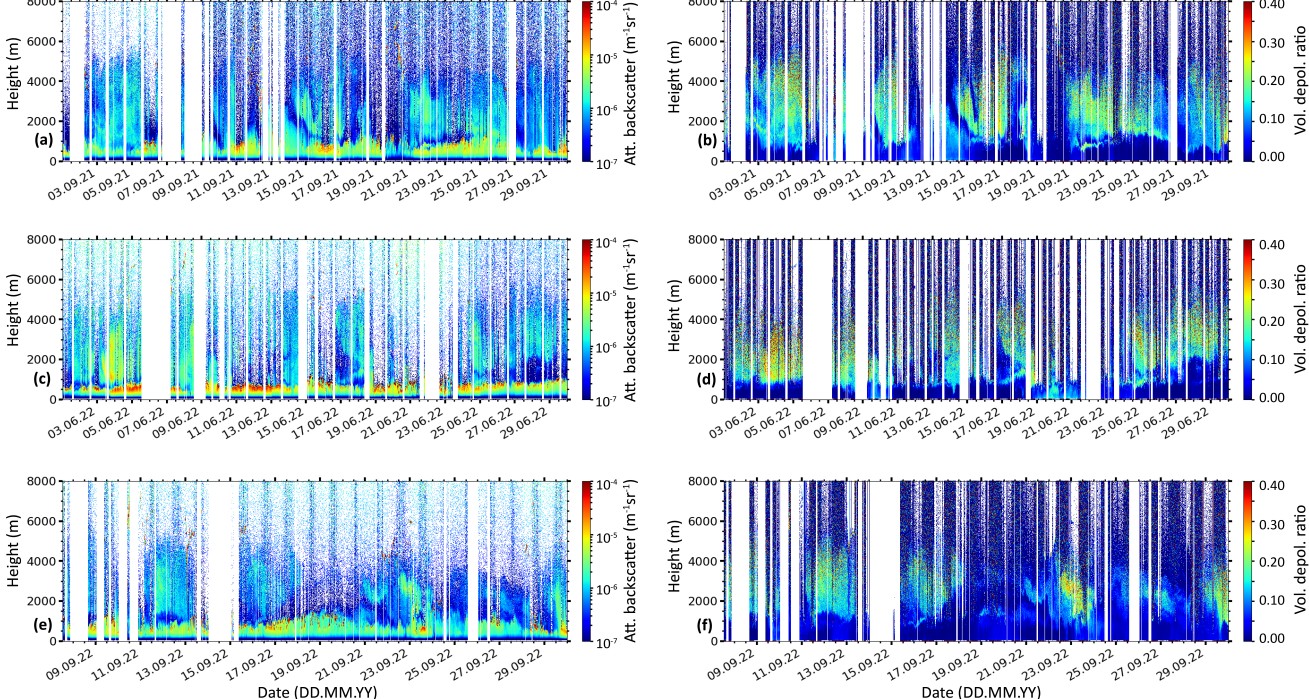

**Figure 2.** Overview of the range-corrected signal at 1064 nm (left column) and volume depolarization ratio at 532 nm (right column) as retrieved from the Polly$^{XT}$ lidar during the ASKOS operations in September 2021 (a, b), June 2022 (c, d), and September 2022 (e, f). No atmospheric data are available during regular depolarization calibration periods, instrument maintenance and during summer daily between 11:30 and 15:00 UTC, due to the location of the instrument with respect to the solar zenith angle (white gaps).

of pure dust is also provided. The well-established LIVAS products have been used in several studies, facilitating the evaluation of climate and aerosol models (Tsikerdekis et al., 2017; Drakaki et al., 2022; Ryder et al., 2024).

The LIVAS dataset used in this study has been derived based on the CALIPSO Level 2 (L2), version 4.5 profiles from June 2006 to June 2023. Mean profiles of aerosol properties have been calculated considering different radii from the ACTRIS ground-based station in Mindelo. More details on the study region are given in Sec. 2.2. The resulting LIVAS dataset is provided with a vertical resolution of 60 m for the atmospheric height range between the surface and 20 km.

## 2.2 Methodology

To assess the degree at which the Polly$^{XT}$ observations at Mindelo can be considered representative for Cal/Val purposes in the region of Cabo Verde, a comparison of the Polly$^{XT}$- and the LIVAS-derived 532-nm backscatter coefficient profiles was performed. First, a study domain had to be selected, which is shown in Fig. 3. It is defined by concentric circles of radii ranging from 20 to 300 km with an incremental step of 20 km from the location of the Polly$^{XT}$ lidar in Mindelo (indicated by a red dot



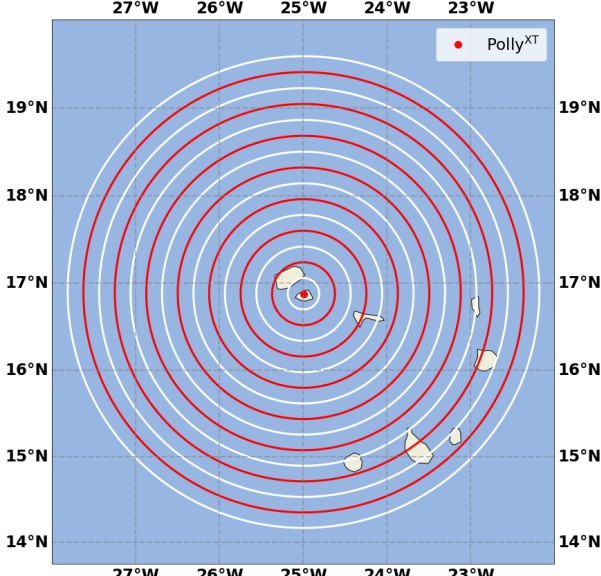

**Figure 3.** Study domain, including the location of the ACTRIS ground-based station in Mindelo, Cabo Verde (red dot). The radii around the station range from 20 to 300 km, with an incremental step of 20 km.

in Fig. 3). The radii were selected such as to cover different satellite overpass scenarios, ranging from a very close overpass (radius less than 20 km) to a rather distant overpass (radius greater than 100 and up to 300 km). Additionally, a maximum radius of 300 km was chosen as to include the southernmost islands of the archipelago (Sotavento Islands).

Second, both datasets were prepared for the comparison. For each ASKOS month, all available automatically-derived Polly$^{\text{XT}}$ cloud-free backscatter coefficient profiles at 532 nm were collected and averaged, resulting in a single monthly-averaged height-resolved aerosol-only profile. Similarly for LIVAS, all CALIPSO overpasses that were within the study domain were automatically aggregated in the LIVAS dataset (see also Sec. 2.1). Then, monthly-mean profiles of the 532-nm backscatter coefficient were calculated taking into consideration all the data within the radii around the ground-based station. It should be noted that the number of CALIPSO overpasses is not the same as the number of profiles examined for a given LIVAS grid cell, since an overpass contains multiple profiles.

To prove that the averaging performed in the aforementioned backscatter coefficient profiles from both datasets is not introducing any biases in our study, three case studies with overpasses at different distances from the ground-based station and varying atmospheric conditions have been analysed and are presented in detail below. The case studies were specifically selected to explore the impact of the vertical and horizontal variability of the target.

## 2.3 Air mass source attribution

The Hybrid Single-Particle Lagrangian Integrated Trajectory model (HYSPLIT; Stein et al., 2015) was used to identify the origin of the air masses that were observed above Mindelo during the ASKOS intensive periods. For each month, two 168-h





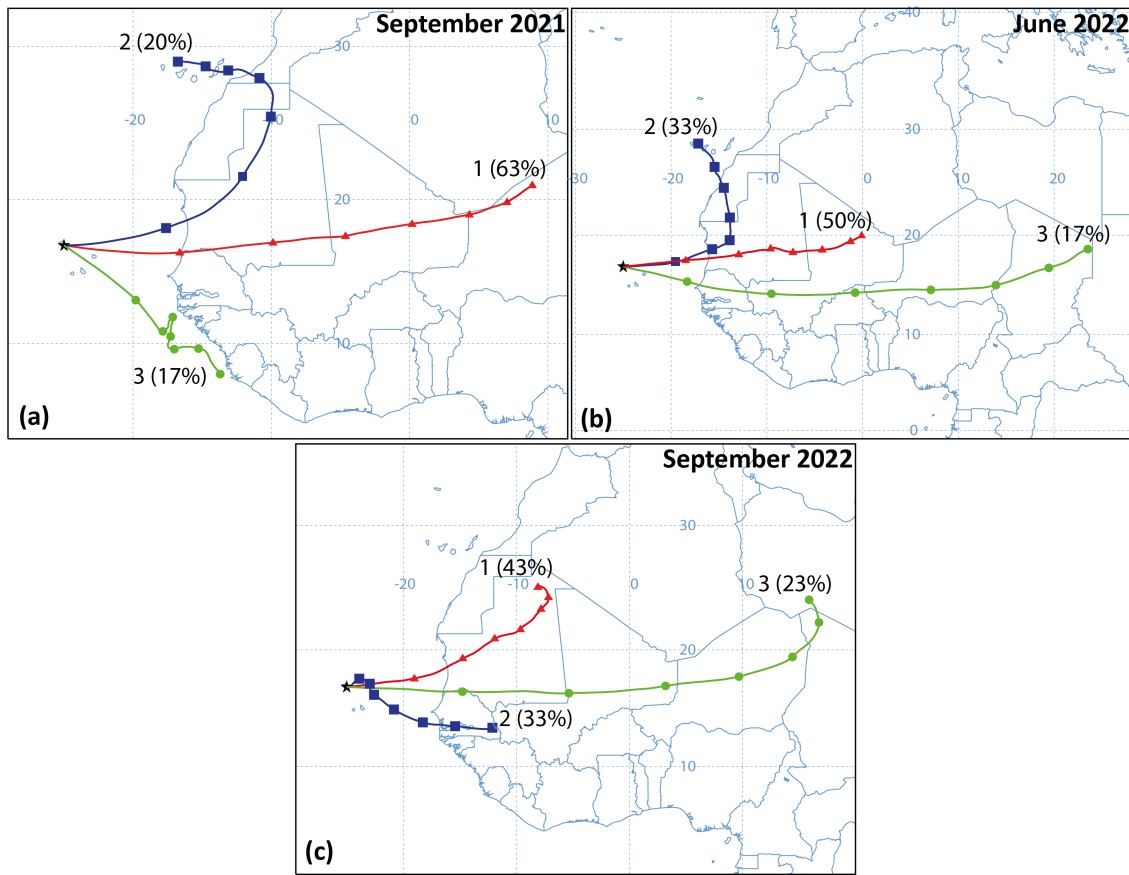

**Figure 4.** HYSPLIT cluster analysis for September 2021 (a), June 2022 (b) and September 2022 (c) based on daily 168-h backward trajectories arriving at Mindelo (black star) at 3 km altitude. The cluster number, along with the percentage of the mean trajectories is indicated.

(i.e., 7-day) backward trajectories were calculated per day (at 06:00 and 18:00 UTC) arriving at the Mindelo station at an altitude of 3 km a.g.l. The altitude was chosen since, based on the observations of Fig. 2, in most cases it coincides with the center of the lofted aerosol layers and at the same time captures lofted layers that didn't reach much higher altitudes. A cluster analysis was then performed on a monthly basis, as shown in Fig. 4. The analysis was performed for 3, 4 and 5 clusters, however, only the results based on 3 classes are shown here, as this number of clusters was found to represent the main sources of the air masses sufficiently.

The mean trajectories (expressed in %) of each cluster are shown in Fig. 4 for September 2021, June 2022 and September 2022 (panels a, b and c, respectively). During September 2021 (Fig. 4a), the majority of the air masses originated from Western Africa (63 %), a known source of mineral dust and biomass-burning aerosol (Tesche et al., 2009). The second most-dominant air mass cluster originated from North Africa (20 %) and crossed parts of Western Africa. A small fraction (17 %) of the air





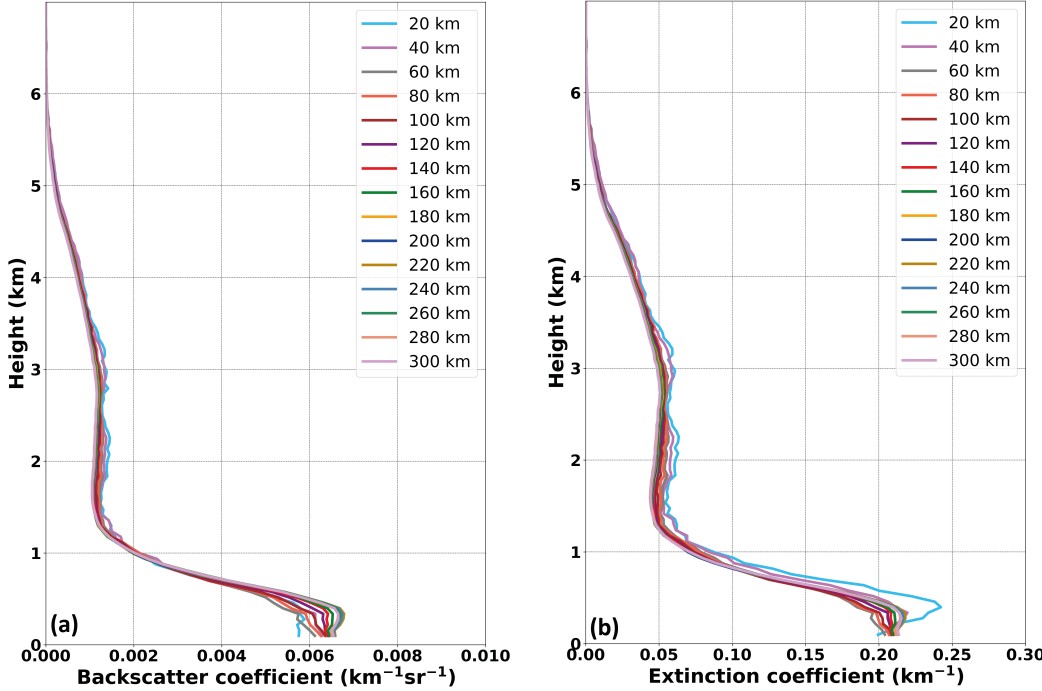

**Figure 5.** 17-year average (2006–2023) backscatter (a) and extinction (b) coefficients (532 nm) from LIVAS (total aerosol product), as a function of distance from the Mindelo station.

masses arriving at Mindelo in September 2021 originated from the North Atlantic Ocean coastline of Sierra Leone, Guinea and Guinea-Bissau.

The cluster analysis of the backward trajectories revealed a similar pattern for June and September 2022 (Fig. 4b and c, respectively). Most of the air masses originated from Western Africa (50 and 43 % for the months of June and September, respectively).

The observations of the lofted aerosol layers from Polly$^{XT}$ (Fig. 2) in combination with the backtrajectory information revealed that the air masses were mostly directly advected from Western Africa, suggest that the observed aerosol particles were either pure dust or a mixture of dust with biomass-burning and sea-salt particles.

## 3 Results

### 3.1 Long-term optical profiles from LIVAS from 2006 to 2023

Over the course of 17 years, from June 2006 to June 2023, a total number of 232909 CALIPSO profiles were acquired within a radius of 300 km from Mindelo. Less than half of the aforementioned profiles were flagged as cloud-free (approximately 46 %, i.e., 108034 profiles). The exact number of available profiles per radius (total and cloud-free) is given in Table A1.



The cloud-free optical profiles were first classified according to their distance to the Mindelo station and then averaged per
radius and are presented in Fig. 5. Overall, the backscatter and extinction profiles per radius are very similar, independently of
the distance to the station, revealing that the atmosphere around Mindelo is comparably homogeneous. Both the backscatter
(Fig. 5a) and the extinction coefficients (Fig. 5b) reach maximum values within the PBL. The average PBL top height appears
to be around 0.5 km. Above that height and up to approximately 1.3 km, both optical parameters decrease, indicating the
transitioning zone between the PBL and the lofted aerosol layers. Between 1.3 and 5 km, both the backscatter and the extinction
coefficients reach two secondary maxima, which is indicative of the presence of one or more lofted aerosol layers.

The backscatter coefficient maximum values occurring within the PBL are increasing with increasing distance from the
station. The same holds for the mean extinction coefficient, but for radii greater than 40 km. However, this aforementioned
relationship should be interpreted cautiously, as there might be an effect of the increased sample size, leading to a decreased
variability within the sampling distribution. As there are much less profiles used for the mean profiles for radii below 40 km
compared to larger radii (see Table A1), the average profiles for small radii are much more dominated by single events compared
to the profiles for larger radii. The mean extinction coefficient observed at distances smaller than 40 km from the Mindelo
station exhibits the highest values. The maximum mean extinction coefficient for radii less or equal to 20 km could be associated
to several reasons including the predominance of fine-mode aerosol particles of anthropogenic origin from the island of São
Vicente and, thus, the potentially wrong lidar ratio assignment by the CALIPSO aerosol subtype selection scheme (Kim et al.,
2018). It should be noted that in the following section (Sec. 3.2), the monthly variability of the extinction coefficient as a
function of distance from the Mindelo station will not be examined, solely due to the fact that it is influenced by the aerosol
subtype selection scheme (Kim et al., 2018), a typical problem for elastic backscatter lidars. Therefore, we consider only
backscatter profiles.

## 3.2 Monthly comparison with Polly$^{XT}$ profiles

A total number of 1241, 1261 and 1306 LIVAS profiles were identified within the 300 km radius from the Mindelo station
during September 2021, June 2022 and September 2022, respectively. Approximately 53, 45 and 63 % of the aforementioned
profiles (for the months of September 2021, June 2022 and September 2022, respectively) were cloud-free and used further for
the comparison with the cloud-free Polly$^{XT}$ profiles. Similarly to Sec. 3.1, the cloud-free profiles were classified according to
their distance to the Mindelo station and respective statistics such as the mean, median and associated erros were calculated.
The exact number of profiles from both LIVAS and Polly$^{XT}$ datasets are provided in Tables A2 and B1, respectively. Only the
backscatter coefficient at 532 nm is considered here, since it is a property less affected by the a-priori choice of the lidar ratio.

Figure 6 shows the monthly mean and median backscatter coefficient comparison for the ASKOS months. For September
2021, both the mean and median LIVAS backscatter coefficient profiles at 532 nm (Fig. 6a and d, respectively) reach their
maximum values at an altitude of approximately 0.5 km, regardless of the distance of the profiles to the station. The extremely
high values of the mean LIVAS backscatter coefficient for radii less than 60 km can be associated with potential cloud contami-
nation, or outliers. Given the high values of the median LIVAS backscatter coefficient for a radius less than 20 km (Fig. 6d), we
can conclude that most likely some of the six (Table A2) LIVAS profiles were in fact cloud contaminated. The Polly$^{XT}$-derived





**Figure 6.** Monthly mean (top row) and median (bottom row) backscatter coefficient at 532 nm for September 2021 (a, d), June 2022 (b, e) and September 2022 (c, f) from the LIVAS dataset for different radius around Mindelo (colored solid lines) and as derived from the Polly$^{XT}$ lidar system (solid black line). The uncertainties of the Polly$^{XT}$ data correspond to the standard deviation.





mean and median backscatter coefficient (black line in Fig. 6a and d) reach maximum values also within the PBL, but at a lower altitude of 0.2 km and exhibit slightly lower mean and median values ranging between 0.001 and 0.009 and 0.003 and 0.006 km$^{-1}$sr$^{-1}$, respectively. The high standard deviation and median absolute deviation values accompanying the Polly$^{XT}$ data below 1 km are simply an indicator for the strong variability of aerosol load expected in the PBL.

Similar pattern for the PBL is observed in September 2022 as well (Fig. 6c and f). The mean and median LIVAS backscatter coefficient reach values up to 0.007 and around 0.005 km$^{-1}$sr$^{-1}$, respectively at an altitude of approximately 0.5 km for a maximum radius of 160 km from Mindelo. The corresponding mean and median Polly$^{XT}$ backscatter coefficient reach their maximum values of 0.0045 and 0.0035 km$^{-1}$sr$^{-1}$, respectively at approximately 0.4 km altitude. However, the monthly mean LIVAS backscatter coefficient profiles that correspond to a radius of 40 and 60 km (purple and gray lines in Fig. 6c, respectively) appear to be much lower than the rest of the profiles (note that no profile within 20 km was available during this month). The same behavior, but much less pronounced is also visible in the median profiles (Fig. 6f). This is a sampling size artifact, related to the few number of profiles available at radii less than 60 km.

In June 2022, the backscatter coefficient exhibits a double maxima within the PBL, which is especially pronounced in the mean profiles (Fig. 6b). For the LIVAS data, the first maximum occurs at 0.3 km altitude and the second one approximately at 0.5 km. The maximum occurrence of the Polly$^{XT}$ data coincides with the first maximum from the LIVAS dataset, however, the occurrence of the second maximum is slightly shifted for the Polly$^{XT}$ data, occurring at an altitude of 0.8 km. Given that the aforementioned maxima are less pronounced in the median profiles (Fig. 6e) it can be concluded that either they can be associated with artifacts in the cloud screening routines of both datasets or that they are occurring due to the different spatiotemporal resolutions of the datasets or that they are associated with different events. In addition, it should be noted that for June 2022 the median profiles of the LIVAS backscatter coefficient at 532 nm at altitudes between approximately 1 and 1.8 km were zero. This happens because at this altitude there were several CALIPSO aerosol profiles with features classified as clear air. By default in the LIVAS production, these clean air features are set to zero. Polly$^{XT}$ does not confirm this, as a particle backscatter is observed. Thus, we conclude that CALIPSO was too weak at the end of its lifetime to resolve these aerosol layers close to the ground.

The comparison results are very satisfactory for the altitude range of 1.5 to 5 km, at which lofted aerosol layers typically corresponding to the Saharan Air Layer (SAL), are frequently observed. Especially for June 2022, the agreement between the two datasets was excellent (Fig. 6b and e from 2 km onwards). In September 2021 (Fig. 6a), the Polly$^{XT}$-derived mean backscatter coefficient compared well against the LIVAS profiles, with the exception of profiles being less than 40 km away from Mindelo (Fig. 6b, light blue and purple lines, respectively). For those distances (20 and 40 km), the LIVAS-derived backscatter coefficient is severely overestimated compared to the rest of the LIVAS data (corresponding to distances grater than 40 km) and to the Polly$^{XT}$ data, due to the few profiles (6 and 14 cloud-free profiles, respectively). This pattern is not visible in the median profiles (Fig. 6d), with the exception of the profiles that are within 20 km from the ground-based station. The monthly mean and median backscatter coefficient at this altitude range appears to be underestimated by the LIVAS dataset for September 2022 (Fig. 6c and f, respectively) compared to the one derived from the Polly$^{XT}$ lidar. This underestimation is not linked to the sampling size, as it occurs regardless of the distance to the Mindelo station. It is probably related to the





end-of-lifetime performance of CALIOP (Tackett et al., 2025). Additionally, the two LIVAS backscatter-coefficient maxima occurring at approximately 1.5 and 2.3 km altitude for radii between 40-60 km are cloud-screening-related artifacts.

From the presented monthly comparison of the LIVAS dataset with the Polly$^{XT}$ dataset, several conclusions can be drawn with respect to validation strategies already. Firstly, the importance of the sampling size should be highlighted. The usage of more spaceborne-derived profiles over an extended radius should be prioritized since it is increasing the representativeness of the samples for the mean monthly atmospheric conditions. In addition, our analysis indicates that mean long-term ground-based observations perform better over the respective median values for the validation of mean profiles derived from satellite

observations. The mean backscatter coefficient profiles (Fig. 6a, b and c) captured better the aerosol vertical distribution, while the median profiles (Fig. 6d, e and f) underestimated the aerosol layer top height.

### 3.3    Case studies

To assess the representativeness of the station in terms of direct single profile comparison, three case studies have been examined from the three intensive ASKOS months. The cases were selected based primarily on the proximity of the CALIPSO

overpass to the ACTRIS ground-based station (ranging from as close as 6.50 km and up to 129.10 km) and secondarily on the atmospheric conditions that prevailed during that day. The maximum distance of the CALIPSO overpass to the ground-based station was chosen such as to exceed the radius threshold of 100 km, which is commonly used by the Cal/Val communities for the validation of spaceborne profilers (Baars et al., 2023; Amiridis et al., 2025). The first case study, on 5 September 2021 (Sect. 3.3.1), examines a close overpass of CALIPSO around the ground-based station at Mindelo with a distance of 6.5 km,

while a homogeneous dust layer was present. The second case is from 17 June 2022 (Sect. 3.3.2) and the distance between the CALIPSO ground track and the ground site was 86.5 km. In contrast to the first case, the lofted aerosol layers on 17 June 2022 contained dusty mixtures. In the last case study, 11 September 2022, the distance between the CALIPSO overpass and the ground-based station is 129.1 km and similarly to the first case, here, a lofted layer containing desert dust was observed too.

### 3.3.1    5 September 2021

The first closest CALIPSO overpass examined here occurred on 5 September 2021 at around 04:16 UTC (nighttime) and it was as close as 6.5 km from the ground-based station at Mindelo. On the same day, the ground-based lidar observations revealed a rather dense and stable dust layer over Mindelo with cloud-free conditions during the overpass (Fig. 7). The vertically-resolved attenuated backscatter coefficient at 1064 nm and volume depolarization ratio at 532 nm as derived from the Polly$^{XT}$ lidar between 00:00 and 11:00 UTC are shown in Fig. 7a and b, respectively. The PBL, extending up to an altitude of 600–800 m, is

characterized by moderate backscatter coefficient and lidar ratio values (approximately 50–60 sr at 355 and 532 nm, not shown here) and low particle linear depolarization ratio values (around 10 % at 532 nm, not shown here), indicating the presence of an aerosol mixture containing mostly spherical marine particles mixed with non-spherical desert dust particles. Between 05:30 and 08:00 UTC, low-level clouds formed at the top of the PBL, which caused signal attenuation above the cloud base. Above the PBL and up to almost 6 km altitude, the desert dust layer is located, as indicated by the particle linear depolarization ratio

values that reached values up to 30% at 532 nm.



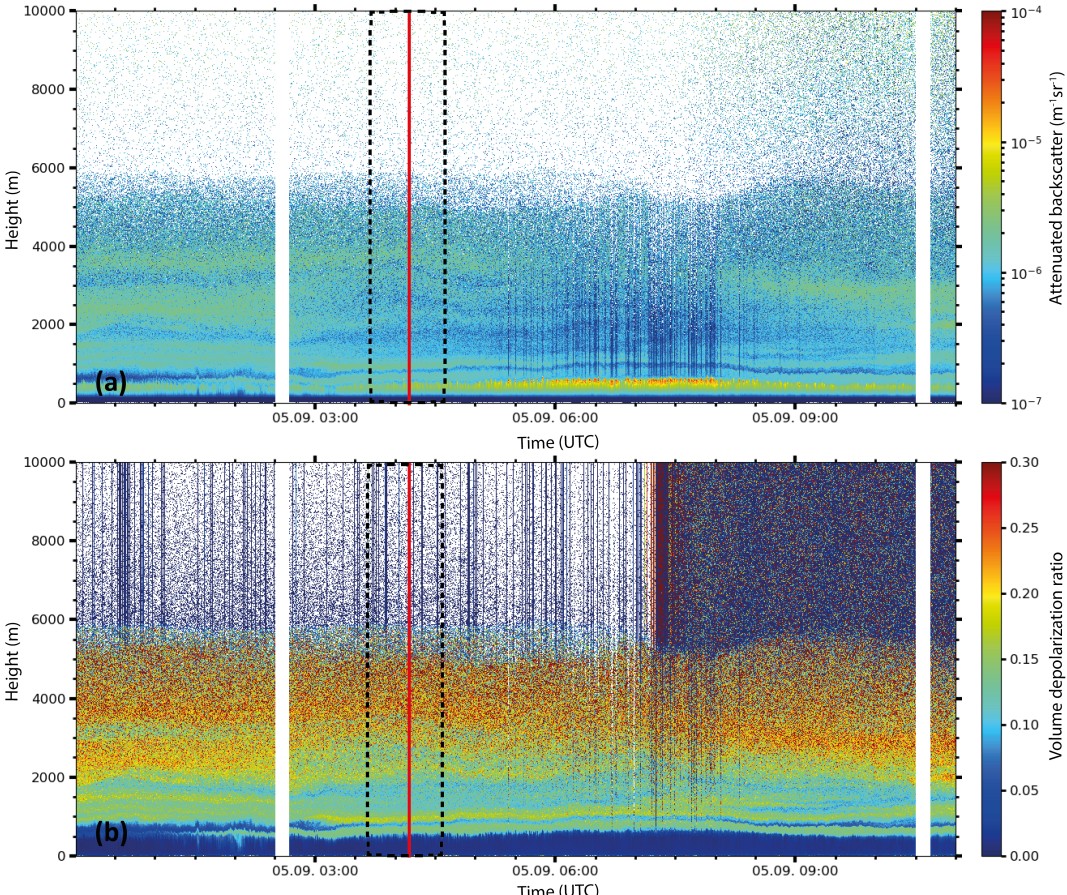

**Figure 7.** Overview of the atmospheric conditions in terms of (a) range-corrected signal at 1064 nm and (b) linear volume depolarization ratio at 532 nm at Mindelo, Cabo Verde, on 5 September 2021 between 00:00 and 11:00 UTC. The red line indicates the time of the CALIPSO overpass, while the black dashed lines indicate the time interval used for the Polly$^{XT}$ retrievals. No data are available during regular depolarization calibration periods (white bars).

The dust layer on the 5 September 2021 was also captured by the MSG's (Meteosat Second Generation) SEVIRI (Spinning Enhanced Visible and Infrared Imager) Dust product and is depicted in Fig. 8 (at 04:00 UTC, a few minutes prior to the overpass). The product is an RGB (Red, Green, Blue) composite based on infrared channel data (IR8.7, IR10.8 and IR12.0) and has been created to monitor the evolution of dust storms. Pink to violet colors indicate dust. It can be seen that the dust layer observed above Mindelo appears to have a greater extent, covering all the Cabo Verde islands and the archipelago around them for a radius of at least 600 km, exhibiting high spatial homogeneity.

The Polly$^{XT}$ and LIVAS optical profile comparison is shown in Fig. 9. The Polly$^{XT}$-derived optical properties were retrieved with the Raman method for an 1-h cloud-free interval (03:41–04:40 UTC), coinciding with the CALIPSO overpass at 04:16 UTC. While the Polly$^{XT}$ products are available at three wavelengths (355, 532 and 1064 nm), for the purposes of this



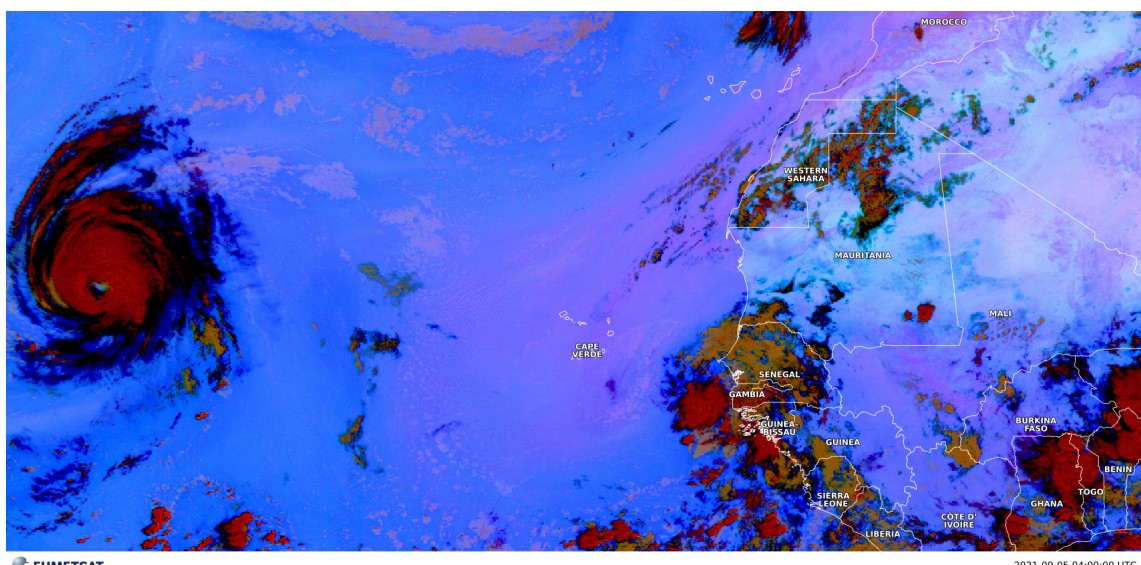

**Figure 8.** MSG-SEVIRI Dust product for 5 September 2021 at 04:00 UTC. Pink to violet colors indicate dust layers. Data accessed via the Eumetview platform (https://view.eumetsat.int/; last access: 25 September 2025).

comparison, we focus only on the 532 nm wavelength. Overall, we observe an excellent agreement within the uncertainty range for both the backscatter (Fig. 9a) and the extinction coefficients (Fig. 9b) for the whole profile, with the exception of the altitude range between 1.95 and 2.22 km, where LIVAS profiles were zero (as discussed above, CALIPSO was most probably unable to detect such aerosol layers close to the ground; Tackett et al., 2025). The monthly comparisons (see Sect.3.2) revealed deviations in the two datasets for the PBL (altitudes less than 1 km), however, this is not the case for this specific case study, which exhibits very homogeneous atmospheric conditions and takes into account the closest CALIPSO overpass, with a distance of only 6.5 km from the ground-based station.

Above the PBL, we observe a very good agreement at altitudes between approximately 3 and 4.5 km, which confirms the findings of the monthly comparisons (Sect.3.2). An excellent agreement is observed at the same altitudes for the extinction coefficient, which clearly demonstrates the retrieval improvements induced by the revised lidar ratio selection algorithm (Kim et al., 2018) and the correct aerosol typing performed for this case of almost pure Saharan dust.

### 3.3.2  17 June 2022

The second case examined here corresponds to 17 June 2022. For this case, the CALIPSO overpass occurred at approximately 16:16 UTC (daytime) and the minimum distance from the ground-based station was 86.54 km.

Figure 10 depicts the atmospheric conditions over Mindelo for that day. Based on the attenuated backscatter coefficient at 1064 nm (Fig. 10, top panel), we can conclude that the PBL was as usually shallow, reaching altitudes up to 750 m and containing marine particles, as indicated by the low values of the depolarization ratio (Fig. 10, bottom panel). Above the



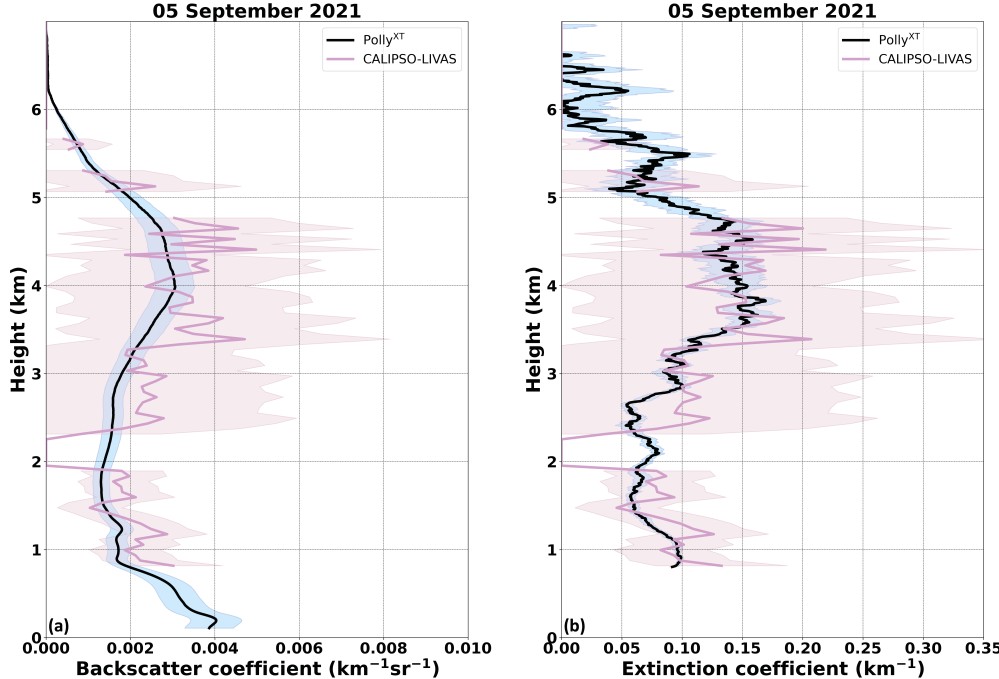

**Figure 9.** Backscatter (a) and extinction (b) coefficient profiles at 532 nm as derived from the Polly$^{XT}$ Raman lidar (black lines) and from the CALIPSO-LIVAS database (lila lines) for the case of 5 September 2021. The Polly$^{XT}$ and CALIPSO-LIVAS associated uncertainties are indicated with blue and pink shaded areas, respectively.

PBL and up to an altitude of 2 km, several aerosol layers were present, stack on top of one another. These aerosol layers are characterized by moderate to low values of attenuated backscatter coefficient and moderate depolarization values, indicating the presence of a mixture of dust and marine aerosol or even dehumidified marine aerosols (Haarig et al., 2017; Bohlmann et al., 2018). Low-level clouds formed at the top of the PBL occasionally throughout the day, but especially during nighttime hours. Above 2 km, the predominance of dust-dominated aerosol is evident in a geometrically thick aerosol layer that extends up to 5 km in altitude. The layer is also visible in the SEVIRI Dust product and it appears to be rather spatially homogeneous (Fig. C1). Mid-altitude clouds, forming at the top of the dust layer, were observed between 03:00 and 06:00 UTC and between 21:30 and 23:00 UTC.

The closest available Raman-based optical parameters from the Polly$^{XT}$ lidar were retrieved automatically by the PPC, for the time period 20:24–21:01 UTC, and the comparison between the Polly$^{XT}$ and CALIPSO-LIVAS backscatter and extinction profiles is shown in Fig. 11. Additionally, since it is a daytime overpass, the Klett-based optical parameters from 16:34–17:34 UTC (also automatically retrieved by the PPC with a pre-set lidar ratio of 40 sr) are also examined and shown in Fig. 11 (the time period for the Klett retrieval is not indicated in Fig. 10). A good agreement is observed for the backscatter coefficient (at 532 nm and for both Raman and Klett retrievals) at altitudes between 2.8–5 km. The difference between the aerosol profiles



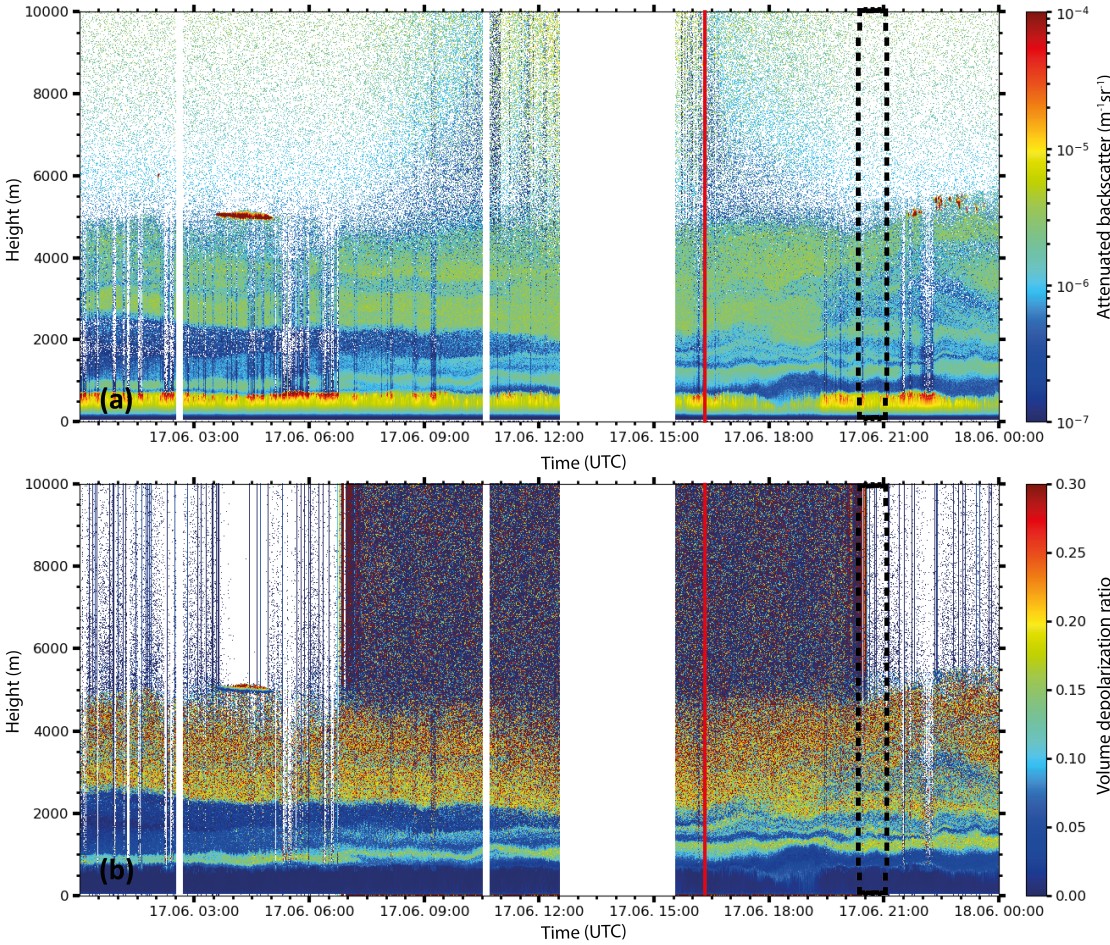

**Figure 10.** Similarly to Fig. 7, but for 17 June 2022 between 00:00 and 24:00 UTC. No data (white bars) are available during regular depolarization calibration periods and between 12:30 and 15:30 UTC, due to the location of the instrument with respect to the solar zenith angle.

during daytime and nighttime is also an indication of the variability of the optical properties of the dust plume on this day. Nevertheless, considering the uncertainties, both profiles (backscatter and extinction) agree well with the LIVAS profile. Larger discrepancies were observed within the PBL and up to 2.8 km (Fig. 11a). The geometrically thin aerosol layers above the PBL are not captured by CALIPSO (Tackett et al., 2025), instead, they were classified as clean air and, hence, set to 0 by the LIVAS production rules. The differences between the Raman and Klett solutions are caused by the fixed lidar ratio of 40 sr, which is used for the automatic processing and is valid for dust but not for marine aerosol and its mixtures (see Floutsi et al., 2023) and the incomplete overlap of the lidar below 900 m, which is not corrected in the automatic processing. Thus, the deviation between the ground-based profiles are due to methodological issues and not due to a stronger temporal inhomogeneity. Given the facts discussed above, the observed discrepancies between the ground-based lidar and CALIOP within the PBL could be attributed



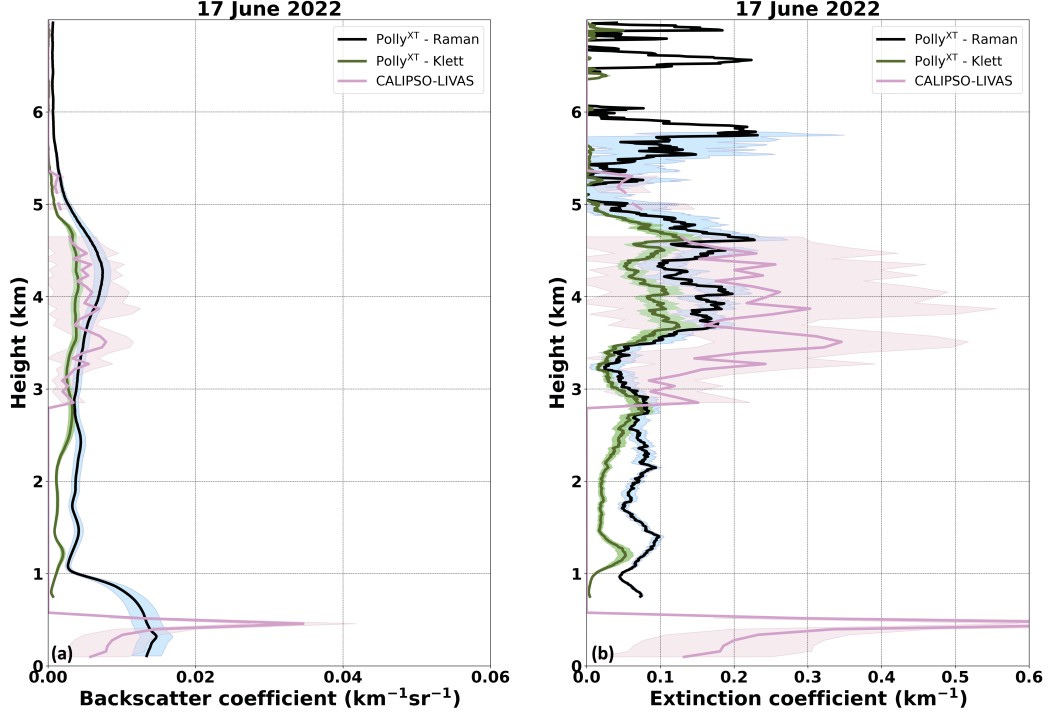

**Figure 11.** Same as Fig. 9, but for the case of 17 June 2022. Additionally, the Klett-retrieved profiles are shown (dark olive green solid line), along with the associated uncertainties (green shaded area). Time intervals for the Raman and Klett retrievals are 20:24–21:01 and 16:34–17:34 UTC, respectively.

to spatial inhomogeneity and most likely to a potential cloud contamination within the LIVAS database. In Fig.10, a low-level cloud can be identified at 500 m shortly after 16:00 UTC (exactly at the altitude at which the CALIPSO-LIVAS backscatter coefficient maximum occurs). The extinction coefficient from both data sources is shown in Fig. 11b. Before discussing the extinction coefficient comparison, it should be noted that the Klett-based extinction coefficient has been calculated by using the nighttime lidar ratio measured from 20:24–21:01 UTC. The extinction coefficient derived from the CALIPSO-LIVAS data

is slightly overestimating the aerosol load at the altitude range of 3–4.5 km, but agrees within the given uncertainties with both the Raman- and the Klett-based Polly$^{XT}$ retrievals. Given the differences between the daytime and nighttime profiles from Polly$^{XT}$, but also considering the backscatter comparison, spatiotemporal inhomogeneity seems to be the only reason affecting the comparison. One other plausible explanation for the extinction differences is the lidar ratio used for the CALIPSO extinction retrieval. An aerosol subtype missclassification (e.g., polluted dust instead of dust) might have triggered the selection

of a higher aerosol lidar ratio, thus, resulting in higher extinction coefficient values.



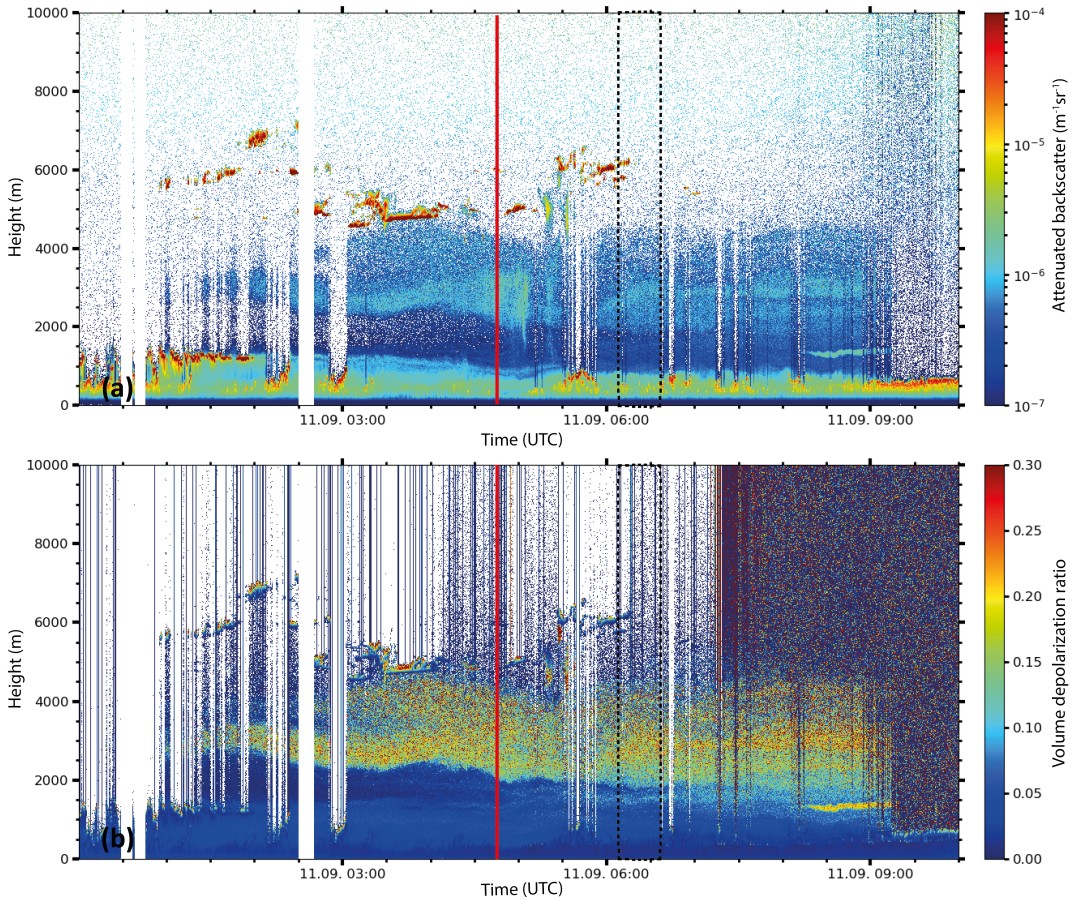

**Figure 12.** Similarly to Fig. 7, but for on 11 September 2022 between 00:00 and 10:00 UTC.

### 3.3.3   11 September 2022

For the third case, an overpass with a minimum distance to the ground-based station that exceeds the commonly-used radius of 100 km was chosen. The selected CALIPSO overpass occurred on 11 September 2022 at approximately 04:44 UTC. The minimum distance between the CALIPSO overpass and the ground-based stations was 129.1 km. Figure 12 depicts the atmospheric conditions over Mindelo. A dust plume reached the Mindelo ground-based station at approximately 01:00 UTC and evolved in a rather dense and stable dust layer at approximately 14:00 UTC (not shown here), as indicated by the attenuated backscatter coefficient at 1064 nm and the volume depolarization ratio at 532 nm (Fig. 12a and b, respectively). A characteristic marine boundary layer is present at altitudes below 2 km with a frequent occurrence of low-level clouds until 13:00 UTC. At the early hours of 11 September 2022, scattered mid-level clouds were also present at altitudes between 5 and 7 km. These clouds are also captured by the SEVIRI Dust product (Fig. C2, yellow colors).





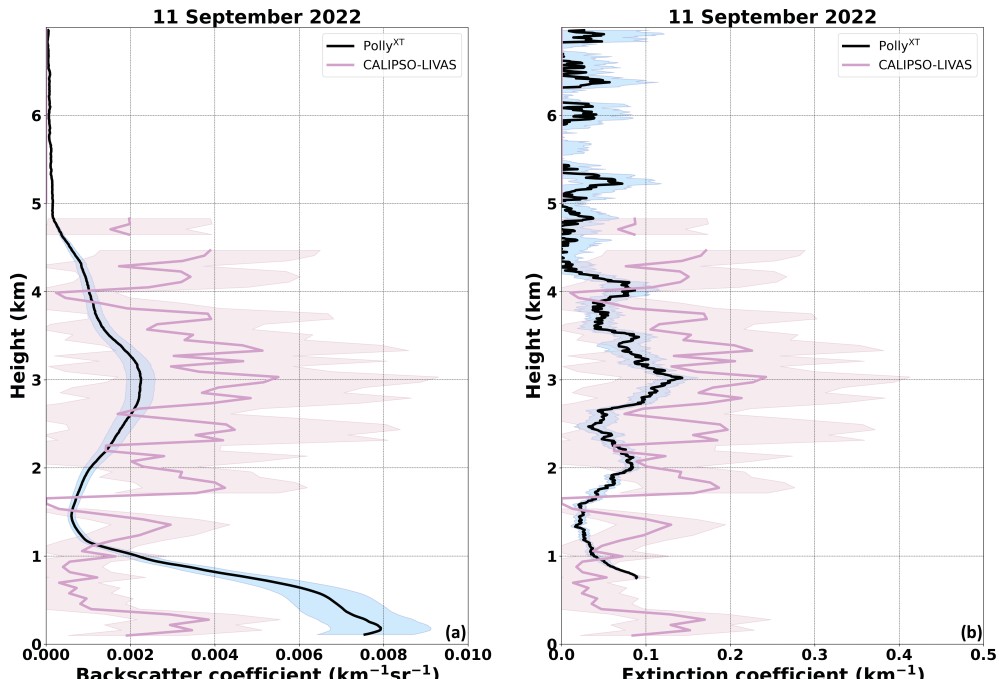

**Figure 13.** Same as Fig. 9, but for the case of 11 September 2022.

The Polly$^{XT}$ and LIVAS backscatter and extinction coefficient profile comparison is shown in Fig. 13a and b, respectively. The closest available, automatically-retrieved Raman-based optical parameters from the Polly$^{XT}$ lidar were from the time period 06:16–06:35 UTC. The CALIPSO-LIVAS backscatter coefficient is underestimating the aerosol load in the PBL, while agreement between the LIVAS and the Polly$^{XT}$ wihin errors at altitudes between 1.7 and 4.5 km is found, however with the ten-
dency of higher values measured by CALIPSO. The same pattern is also observed for the extinction coefficient. The observed discrepancies can be attributed to the spatial inhomogeneity of the dust plume as indicated in Fig. C2, rather than CALIPSO aerosol subtype missclassification. In the SEVIRI Dust product image (Fig. C2), an enhancement of the intensity of the dust towards the north-west from Mindelo is clearly seen, explaining the higher values seen by CALIPSO. Thus, for this case, simple representativeness of the atmospheric conditions between the satellite-based and the ground-based observation cannot
be considered and would require much more sophisticated approaches, such as dispersion modeling.

## 4 Conclusions and Outlook

By utilizing 17 complete years of CALIPSO overpasses, from 2006 till 2022 and within a 300 km radius from the ACTRIS ground-based station in Mindelo, Cabo Verde, the average atmospheric conditions in terms of aerosol were characterized to check to which degree the ground-based fixed lidar measurements are representative, in the context of satellite-based lidar





Cal/Val activities. We utilized continuous multiwavelength Polly$^{XT}$ Raman lidar measurements at Mindelo, to asesss how monthly mean profiles, as well as single case studies, can be used for spaceborne profiles validation. Our results indicate that the representativeness in this specific geographic region is primarily affected by the spatial homogeneity of the observable target and secondary by the co-location of the ground-based and spaceborne instrumentation.

Cabo Verde is well suited for validation of spaceborne aerosol profiles. Measurements conducted in Mindelo by ground-
based lidars like the Polly$^{XT}$ Raman lidar system are shown to be representative for a radius of roughly 300 km in the usually-occurring lofted aerosol layers (in the SAL) when considering the monthly averages. In the free troposphere above the PBL, particles transported from source regions (e.g., desert dust layers) dominate the aerosol load throughout the year. But also the frequently used radius threshold of 100 km seems to be appropriate in the case of Mindelo. The stable atmospheric stratification hinder vertical mixing and lead to homogeneous aerosol layers making it an ideal place for performing validation activities. On
the contrary, the monthly averaged results for the PBL showed higher variability with increasing radius indicating that targets within the PBL, which are mostly originating from local sources, are naturally more susceptible to spatiotemporal variability. The long-term, i.e., monthly analysis results highlight the importance of the sample size. It was shown that monthly averages obtained from less than approximately 40 profiles were not as representative, due to the limited number of samples, the monthly mean values obtained by applying a larger radius and, thus, including much more spaceborne profiles. In conclusion,
for the long-term validation of spaceborne aerosol profiles, it is better to use monthly averaged ground-based lidar profiles rather than single profiles during the overpasses, as for the latter ones, representativeness cannot be guaranteed and may lead to wrong conclusions. If, however, representativeness can be guaranteed, as shown in our first case study, also single profile validation is possible and has its own potentials, but if not, it may lead to wrong conclusions as shown for the third case study. But generally, for the long-term assessment of the spaceborne instrument performance, it seems to be more appropriate
to use monthly-mean profiles and in turn increase the radius around the station to increase sample size and, thus, introduce representativeness. While the results of this study are in the first instance valid for the Cabo Verde region only, it is known that representativeness is challenging for all ground-based stations when validating spaceborne profilers. Thus, similar studies for other geographic regions should be made in the future. Nevertheless, our study highlights that Cabo Verde, with its omnipresent lofted aerosol layer in the SAL, is well-suited for long-term validation of spaceborne aerosol profiles from the ground. At other
ground-based measurement sites, the network approach consisting of multiple stations (e.g., ACTRIS, Baars et al., 2024) might be a good approach to overcome potential representativeness shortcomings and could be investigated in a similar way to the study presented here for Mindelo/ Cabo Verde.

The results of this study underline the importance of the careful evaluation of the spatial and temporal homogeneity, with respect to the validation of aerosol and cloud profilers, e.g., EarthCARE, as also highlighted in a dedicated Best Practice
Document (Amiridis et al., 2025). Currently, several Cal/Val teams from around the globe are actively working on the validation of L1 and L2 EarthCARE products from all instruments. As a common practice, the EarthCARE Cal/Val community uses frequently in the first instance a maximum radius of 100 km around the ground station for validation. According to this study, the maximum radius that can be used depends on the validation approach and determines which criteria to set. For monthly averaged profiles, larger radii might be more appropriate, while for single profile validation, much smaller radii or additional



approaches, such as backtrajectories or dispersion modeling, might be needed to consider the comparison as representative. Therefore, this study can be considered as a pilot study and the methodology could be applied to other stations as well, to investigate the spatial variability of the atmospheric targets in different geographic regions and create effective Cal/Val strategies. Additionally, the study can be used as a tool to assess the spatiotemporal homogeneity of natural and anthropogenic aerosol, which has also air quality implications.

*Code availability.* For the lidar data visualization (Fig. 7, 10 and 12), pyLARDA was used (https://doi.org/10.5281/zenodo.4721311, Bühl et al., 2021).

*Data availability.* The complete ASKOS dataset is available via the ESA Atmospheric Validation Data Centre under (https://doi.org/10.60621/jatac.campaign.2021.2022.caboverde, Amiridis et al., 2023). Quicklooks of the Polly$^{XT}$ lidar products are publicly available at https://polly.tropos.de/, last access: 25 September 2025. The LIVAS dataset and products are available upon request from Konstantinos Rizos
(k.rizos@noa.gr).

**Appendix A: Information on the LIVAS dataset**

Table A1 presents the total number of available LIVAS profiles (including clouds) and cloud-free profiles that were identified at different radii around the ground-based station in Mindelo, Cabo Verde between 2006 and 2023. The number of total and cloud-free LIVAS profiles only for the ASKOS intensive period is shown in Table A2.





**Table A1.** Number of total and cloud-free LIVAS profiles for different radii around Mindelo station for the period 2006–2023.

| Radius (km) | Profiles (#) | Cloud-free profiles (#) |
| --- | --- | --- |
| 20 | 2196 | 964 |
| 40 | 5323 | 2319 |
| 60 | 12546 | 5931 |
| 80 | 18989 | 9015 |
| 100 | 25830 | 12149 |
| 120 | 35177 | 16511 |
| 140 | 48143 | 22665 |
| 160 | 62246 | 29147 |
| 180 | 86767 | 39708 |
| 200 | 106992 | 48645 |
| 220 | 133101 | 60801 |
| 240 | 156047 | 71469 |
| 260 | 177991 | 81545 |
| 280 | 202935 | 93381 |
| 300 | 232909 | 108034 |





**Table A2.** Number of total and cloud-free LIVAS profiles for different radii around Mindelo station during the ASKOS intensive periods.

| Month | September 2021 | | June 2022 | | September 2022 | |
|---|---|---|---|---|---|---|
| Radius (km) | Profiles (#) | Cloud-free profiles (#) | Profiles (#) | Cloud-free profiles (#) | Profiles (#) | Cloud-free profiles (#) |
| 20 | 8 | 6 | 0 | 0 | 0 | 0 |
| 40 | 16 | 14 | 30 | 14 | 13 | 10 |
| 60 | 68 | 64 | 60 | 28 | 22 | 15 |
| 80 | 109 | 96 | 88 | 41 | 66 | 42 |
| 100 | 158 | 127 | 144 | 76 | 97 | 65 |
| 120 | 255 | 168 | 198 | 109 | 168 | 87 |
| 140 | 330 | 196 | 268 | 153 | 230 | 123 |
| 160 | 418 | 241 | 342 | 184 | 343 | 212 |
| 180 | 493 | 272 | 443 | 229 | 436 | 284 |
| 200 | 622 | 325 | 582 | 308 | 573 | 368 |
| 220 | 724 | 367 | 689 | 365 | 683 | 443 |
| 240 | 838 | 411 | 822 | 422 | 835 | 540 |
| 260 | 953 | 476 | 970 | 468 | 970 | 604 |
| 280 | 1082 | 544 | 1129 | 524 | 1143 | 716 |
| 300 | 1241 | 655 | 1261 | 564 | 1306 | 822 |





## Appendix B:  Number of Polly$^{XT}$ profiles at Mindelo


The total number of cloud-free profiles that were derived by the Polly$^{XT}$ lidar measurements during the ASKOS intensive phases in Mindelo, Cabo Verde is shown in Table B1.

**Table B1.** Number of cloud-free optical profiles as derived from the Polly$^{XT}$ lidar measurements at Mindelo for the respective ASKOS months.

| Month | Cloud-free profiles (#) |
|---|---|
| September 2021 | 277 |
| June 2022 | 296 |
| September 2022 | 221 |

## Appendix C:  MSG-SEVIRI Dust product

The MSG-SEVIRI dust product for 17 June 2022 and 11 September 2022 (see Sect. 3.3.2 and 3.3.3, respectively) is shown in

Fig. C1 and C2, respectively. Pink to violet colors indicate dust layers.

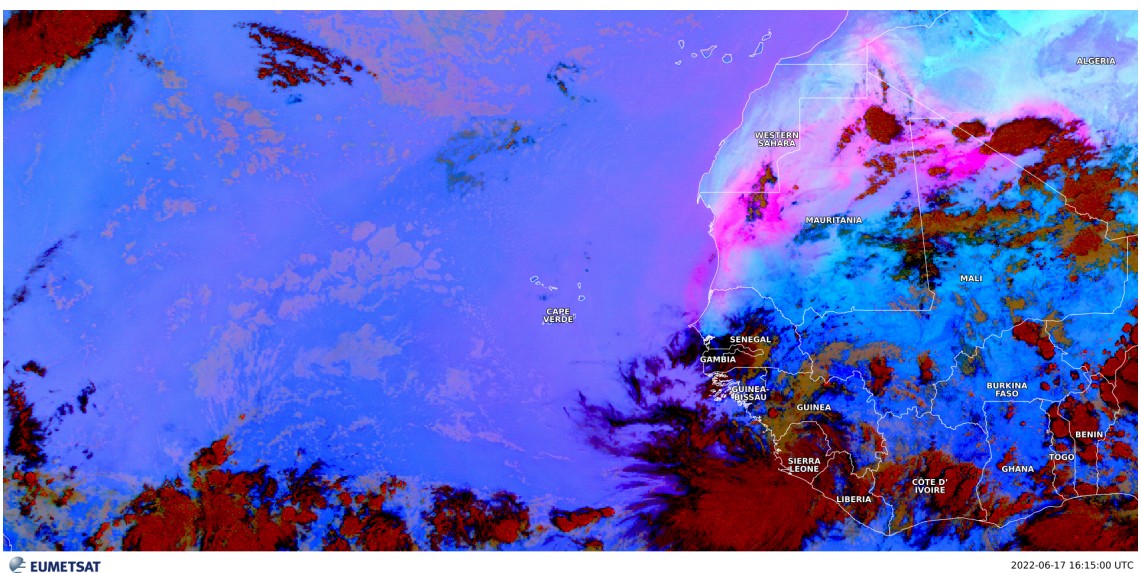

**Figure C1.** Same as Fig. 8, but for 17 June 2022 at 16:15 UTC.



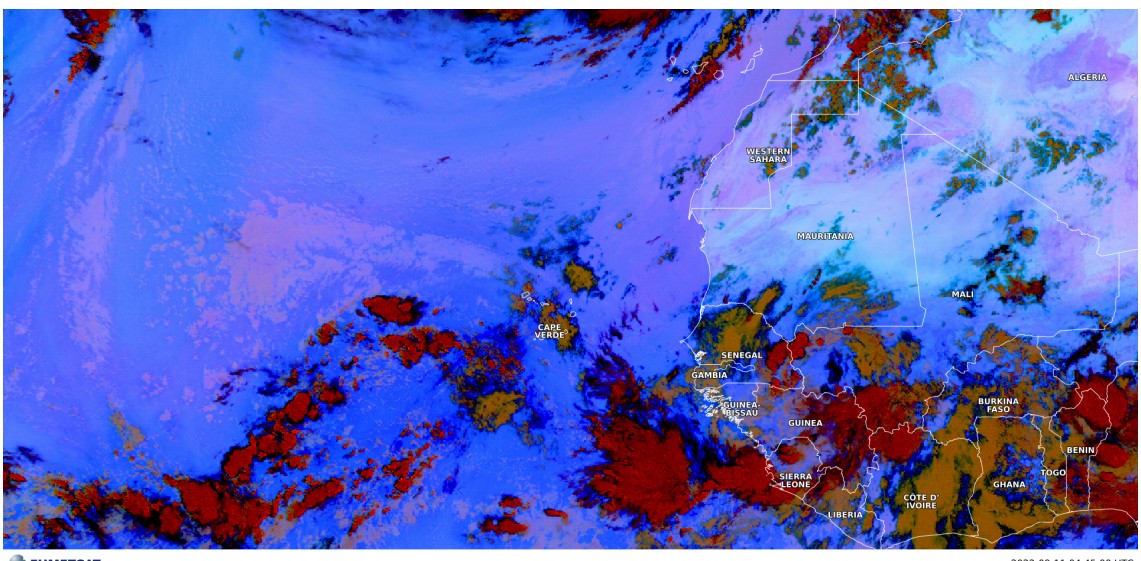

**Figure C2.** Same as Fig. 8, but for 11 September 2022 at 04:45 UTC.

*Author contributions.* AAF analysed the data and drafted the manuscript together with HB. KR derived and provided the LIVAS data. RE, EM, PP, DT, AS, TF, VA and HB were actively involved in the ASKOS campaign. JH performed the backtrajectory analysis. AA, HB and UW contributed to the scientific discussion. All co-authors contributed to the manuscript preparation.

*Competing interests.* Ulla Wandinger and Vassilis Amiridis are members of the editorial board of Atmospheric Measurement Techniques.
The authors have no other competing interests to declare.

*Acknowledgements.* This research has been supported by the German Federal Ministry of Education and Research (BMBF) under the FONA Strategy "Research for Sustainability" (grant no. 01LK2001A), by the German Federal Ministry for Economic Affairs and Energy (BMWi) (grant no. 50EE1721C) and by ESA's project Best practice protocol for validation of Aerosol, Cloud, and Precipitation Profiles (ACPV, no. ESA RFP/3-17843/22/I-NS). The authors wish to thank the OSCM team for their infrastructural support and logistics during the ASKOS
operations and beyond.



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
