# Peer review of "On the representativeness of the ground-based lidar observations for satellite calibration/validation—the example of the archipelago of Cabo Verde"

_EGUsphere, 2025_

## Referee Comment (RC1)

**On the representativeness of the ground-based lidar observations for satellite calibration/validation—the example of the archipelago of Cabo Verde**, Athena Augusta Floutsi, Konstantinos Rizos, Dimitri Trapon, Ronny Engelmann, Dietrich Althausen, Eleni Marinou, Peristera Paschou, Julian Hofer, Emmanouil Proestakis, Henriette Gebauer, Annett Skupin, Albert Ansmann, Thorsten Fehr, Timon Hummel, Rob Koopman, Vassilis Amiridis, Ulla Wandinger, and Holger Baars

**General Comments**

The paper is logically structured, clearly presenting the data, methodology, results from long-term and monthly comparisons, and detailed case studies. The conclusions drawn are well-supported by the evidence, particularly highlighting the conditional representativeness of the Mindelo station.

A key strength of the paper is the clear distinction and recommendation for using monthly averaged profiles over single overpasses for long-term validation. The findings strongly suggest that monthly averaged aerosol profiles are better for validating spaceborne profiles over long times, as representativeness cannot be guaranteed for single overpasses without additional measures.

I would like to offer several constructive observations that may help refine the study and guide future work as follows.

Minor general comments:

- The comparison lacks statistical measures when comparing ground based and satellite derived measurements for monthly analysis.
- The difference between mean and median analysis could be emphasized a bit more in the analysis

**Specific Comments**

- Line 47, please add the Level 1 and Level 2 ATBD references for CALIPSO;
- Line 68, please add the lat/lon information on the JATAC location;
- Line 91, please clarify what "not as part of ACTRIS facility" mean in the context of your cal/val campaign;
- Line 113, please add a reference for aerosol typing and its residence time;
- Line 154, all the aerosol typing analysis throughout ASKOS is based on Floutsi et. al., 2023? Please clarify that this is true for all your analysis. As you are referencing it just for dust aerosol, it is not very clear as this is true for other aerosol types.
- Line 170, please add an example and/or reference for deriving Ångström exponent from optical models.
- Line 175, as above, please add the CALIPSO L2 ATBD reference;
- Line 203, do you have any statistics that supports your choice as a starting altitude that you ran HYSPLIT? How the lofted aerosol layers are calculated from PollyXT system? What is the bin range for each altitude that HYSPLIT considers? Please argue a bit more on how these input value have been chosen.

- Line 225, please clarify the methodology that you used to reach to the statement that "the backscatter and extincion profiles per radius are very similar". What are the statistical criteria you used in your analysis? Please also clarify how you define here that "the atmosphere around Mindelo is comparable homogeneous".
- Fig. 6, Have you used other filtering criteria on both LIVAS and PollyXT data (e.g., associated errors, QA filters) other than cloud screening?
- Line 283, please comment and define the terms "satisfactory" and "severely overestimated" at Line 287. What are the statistical measures used to reach these conclusions?
- Line 314, assuming maximum distance should be less than 100 km between ground based and satellite, the 11 Sept 2022 case is greater than your filtering, please clarify.
- Line 334, please comment and/or argue why the chosen of the ground bsaed retrieval measurements interval was not centered around the satellite overpass.
- Line 441, please add reference to EarthCARE L1 and L2 ATBDs

**Technical Corrections**
- valid throughout document: please be consistent with the acronyms and their explanation. Either use the acronym first and the explanation in parenthesis, e.g., line 36, or full name and the acronym in paranthesis, e.g., line 53.
- Line 96, you end the sentence and start the next one with same word (i.e., EarthCARE), please rephrase.
- Line 307, please clarify how the spatial filtering was done here. The phrase "The maximum distance of the CALIPSO overpass to the ground-based station was chosen such as to exceed the radius threshold of 100 km…", one can understand that the **minimum** distance between the two instruments should be **more** than 100 km.

---

## Referee Comment (RC2)

Review of "On the representativeness of the ground-based lidar observations for satellite calibration/validation—the example of the archipelago of Cabo Verde"

Athena Augusta Floutsi, Konstantinos Rizos, Dimitri Trapon, Ronny Engelmann, Dietrich Althausen, Eleni Marinou, Peristera Paschou, Julian Hofer, Emmanouil Proestakis, Henriette Gebauer, Annett Skupin, Albert Ansmann, Thorsten Fehr, Timon Hummel, Rob Koopman, Vassilis Amiridis, Ulla Wandinger, and Holger Baars

General Comments:

Major Comments:

1. In the Introduction and elsewhere, there was no discussion of or reference to the article by Gimmestad et al., 2017. This article discusses the challenges of validating spacebased lidar using ground-based measurements including random and systematic differences, statistical limitations, averaging, etc. It would be interesting to present the results of the current Cabo Verde study that found monthly averages more useful for validation in light of the results of this previous study. Also, keep this in mind regarding the discussion in the paper in lines 109-113.

    Gimmestad, G., Forrister, H., Grigas, T. et al. Comparisons of aerosol backscatter using satellite and ground lidars: implications for calibrating and validating spaceborne lidar. Sci Rep 7, 42337 (2017). https://doi.org/10.1038/srep42337

2. The summary (line 414) has the statement "Cabo Verde is well suited for validation of spaceborne aerosol profiles." Later (line 418) is the statement "The stable atmospheric stratification hinder vertical mixing and lead to homogeneous aerosol layers making it an ideal place for performing validation activities." The examples presented in the paper show that, although these statements are true for aerosols above the MBL, there is greater difficulty for using profiles within the MBL. The next sentence in the summary (line 420) gives some indication of this "On the contrary, the monthly averaged results for the PBL showed higher variability with increasing radius indicating that targets within the PBL, which are mostly originating from local sources, are naturally more susceptible to spatiotemporal variability." My suggestion is to provide greater clarity regarding this point so the statement in line 414 could be modified to "Cabo Verde is well suited for validation of spaceborne aerosol profiles, in particular for aerosol layers above the MBL." Likewise a similar statement in the abstract would be helpful.

Specific Comments:

1. Line 41. add "typically" so the sentence reads "The lidar ratio (extinction-to-backscatter ratio) typically had to be assumed to enable…"
2. Line 45. While the statement is true, the sentence makes it sound like a more capable lidar (e.g., HSRL, Raman) that can directly measure the lidar ratio has little or no need for cal/val. I suggest changing the sentence to read something like "Because of this, validation of CALIOP's products was particularly necessary and so was performed by means of direct comparisons with ground-based and airborne measurements."

3. Lines 52-54. This paragraph is misleading and unbalanced. There are three sentences describing a single airborne lidar mission and publication for CALIPSO Validation (i.e. McGill et al., 2007) and only a single sentence describing the extensive work and numerous publications associated with CALIPSO validation via airborne HSRL measurements. I suggest modifying this single sentence to be "Throughout the mission's lifetime, extensive collocated underflights (see https://www-air.larc.nasa.gov/missions/calipso-hsrl-underflights/index.html) of the NASA Langley Research Center airborne high-spectral-resolution lidars (HSRLs) took place to assess CALIOP's calibration accuracy (Powell et al., 2009; Rogers et al., 2011; Kar et al., 2018; Vaughan et al., 2019), aerosol classification and lidar ratio algorithm (Omar et al., 2009; Burton et al., 2013), CALIOP aerosol lidar ratio and aerosol optical depth retrievals (Josset et al., 2011; Rogers et al., 2014; Ryan et al., 2024; Ferrare et al., 2024), and CALIOP retrievals of aerosol extinction profiles (McPherson et al., 2010; Burton et al., 2010; McPherson and Reagan, 2016; Painemal et al., 2019)." As per major comment 1, this also highlights the utility of airborne measurements in relation to ground-based measurements.

The additional references mentioned above are:

Burton, S. P., Ferrare, R. A., Hostetler, C. A., Hair, J. W., Kittaka, C., Vaughan, M. A., Obland, M. D., Rogers, R. R., Cook, A. L., Harper, D. B., and Remer, L. A.: Using airborne high spectral resolution lidar data to evaluate combined active plus passive retrievals of aerosol extinction profiles, J. Geophys. Res.-Atmos., 115, D00H15, https://doi.org/10.1029/2009jd012130, 2010.

Burton, S. P., Ferrare, R. A., Vaughan, M. A., Omar, A. H., Rogers, R. R., Hostetler, C. A., and Hair, J. W.: Aerosol classification from airborne HSRL and comparisons with the CALIPSO vertical feature mask, Atmos. Meas. Tech., 6, 1397–1412, https://doi.org/10.5194/amt-6-1397-2013, 2013.

Ferrare R, Hair J, Hostetler C, Shingler T, Burton SP, Fenn M, Clayton M, Scarino AJ, Harper D, Seaman S, Cook A, Crosbie E, Winstead E, Ziemba L, Thornhill L, Robinson C, Moore R, Vaughan M, Sorooshian A, Schlosser JS, Liu H, Zhang B, Diskin G, DiGangi J, Nowak J, Choi Y, Zuidema P and Chellappan S (2023) Airborne HSRL-2 measurements of elevated aerosol depolarization associated with non-spherical sea salt. Front. Remote Sens. 4:1143944. doi: 10.3389/frsen.2023.1143944

Josset, D., Rogers, R., Pelon, J., Hu, Y., Liu, Z., Omar, A., and Zhai, P.: CALIPSO lidar ratio retrieval over the ocean, Opt. Express,19, 18696–18706, 2011.

McPherson, C. J., J. A. Reagan, J. Schafer, D. Giles, R. Ferrare, J. Hair, and C. Hostetler (2010), AERONET, airborne HSRL, and CALIPSO aerosol retrievals compared and combined: A case study, J. Geophys. Res., 115, D00H21, doi:10.1029/2009JD012389.

McPherson, C.J. and John A. Reagan "Extension of the constrained ratio approach to aerosol retrievals from elastic-scatter and high spectral resolution lidars," Journal of Applied Remote Sensing 10(3), 036019 (23 August 2016). https://doi.org/10.1117/1.JRS.10.036019

Omar, A., Winker, D. M., Kittaka, C., Vaughan, M., Liu, Z., Hu, Y., Trepte, C. R., Rogers, R. R., Ferrare, R. A., Lee, K-P, Kuehn, R. E., and Hosteler, C. A.: The CALIPSO automated aerosol classification and lidar ratio selection algorithm, J. Atmos. Ocean. Tech., 26, 1994–2014, 2009.

Painemal, D., Clayton, M., Ferrare, R., Burton, S., Josset, D., and Vaughan, M.: Novel aerosol extinction coefficients and lidar ratios over the ocean from CALIPSO–CloudSat: evaluation and global statistics, Atmos. Meas. Tech., 12, 2201–2217, https://doi.org/10.5194/amt-12-2201-2019, 2019.

Powell, K. A., Hostetler, C. A., Liu, Z., Vaughan, M. A., Kuehn, R. A., Hunt, W. H., Lee, K.-P. Trepte, C. R., Rogers, R. R., Young, S. A., and Winker, D. M.: CALIPSO Lidar calibration algorithms. Part I: Nighttime 532 nm parallel channel and 532 nm perpendicular channel, J. Atmos. Ocean. Tech., 26, 2015–2033, https://doi.org/10.1175/2009JTECHA1242.1, 2009.

Rogers, R. R., Vaughan, M. A., Hostetler, C. A., Burton, S. P., Ferrare, R. A., Young, S. A., Hair, J. W., Obland, M. D., Harper, D. B., Cook, A. L., and Winker, D. M.: Looking through the haze: evaluating the CALIPSO level 2 aerosol optical depth using airborne high spectral resolution lidar data, Atmos. Meas. Tech., 7, 4317–4340, https://doi.org/10.5194/amt-7-4317-2014, 2014.

Ryan, R. A., Vaughan, M. A., Rodier, S. D., Tackett, J. L., Reagan, J. A., Ferrare, R. A., Hair, J. W., Smith, J. A., and Getzewich, B. J.: Total column optical depths retrieved from CALIPSO lidar ocean surface backscatter, Atmos. Meas. Tech., 17, 6517–6545, https://doi.org/10.5194/amt-17-6517-2024, 2024

4. Line 129. The inelastic backscatter signals refer to the Raman nitrogen channels, correct? This should be indicated.

5. Line 136. In the discussion of the Polly system, it's not clear the extent to which the measurements discussed in this paper were made during the daytime and/nighttime. Were measurements made during both day and night, and if so, what limitations (if any) are imposed on the daytime measurements? It's not clear the extent to which daytime vs. nighttime measurements were used in the various analyses.

6. Line 149. The recent paper by Shrestha et al. 2026 seems to suggest marine boundary layers can contain dust even though the lidar depolarization is low.
Shrestha, S., Holz, R. E., Marais, W. J., Buckholtz, Z., Razenkov, I., Eloranta, E., Reid, J. S., Elliott, H. E., Lata, N. N., Cheng, Z., China, S., Blades, E., Ortiz, A. D., Chewitt-Lucas, R., Allen, A., Blades, D., Agrawal, R., Reid, E. A., Ruiz-Plancarte, J., Bucholtz, A., Yamaguchi, R., Wang, Q., Eck, T., Lind, E., Pöhlker, M. L., Ault, A. P., and Gaston, C. J.: Transported African Dust in the Lower Marine Atmospheric Boundary Layer is Internally Mixed with Sea Salt Contributing to Increased Hygroscopicity and a Lower Lidar Depolarization Ratio, Atmos. Chem. Phys., 26, 983–999, https://doi.org/10.5194/acp-26-983-2026, 2026.

7. Line 161. At what wavelength is this AOD?

8. Line 188. Cloud-free attenuated or unattenuated backscatter profiles?

9. Line 257. Are the LIVAS profiles supposed to be cloud-free? If these were cloud-contaminated, can the authors provide some information as to how severe a problem is the cloud-contamination?

10. Line 282. When referring to Figure 6, it is not clear whether the profiles and comparisons use daytime and/or nighttime results. How do the comparison results change from day to night?

11. Line 325. What was the lidar ratio of the elevated dust?

12. Figure 7. There is an abrupt transition in the volume depolarization ratio above about 6 km around 0715 UTC.  Why?
13. Line 362.  The uncertainties associated with the Raman retrievals in Figure 11 look fairly small.  If these were daytime retrievals, how much smaller are the uncertainties for nighttime retrievals?  Given how small these uncertainties are in Figure 11, it's not clear why the Klett retrievals were necessary.